# Assessment of the Biocontrol Potential of *Bacillus velezensis* WL–23 against Kiwifruit Canker Caused by *Pseudomonas syringae* pv. *actinidiae*

**DOI:** 10.3390/ijms241411541

**Published:** 2023-07-16

**Authors:** Bingce Wang, Yushan Guo, Xuetang Chen, Jiling Ma, Xia Lei, Weizhen Wang, Youhua Long

**Affiliations:** 1Research Center for Engineering Technology of Kiwifruit, College of Agriculture, Guizhou University, Guiyang 550025, China; wbcgzu@126.com (B.W.); 15185672765@163.com (Y.G.); chenxt951231@126.com (X.C.); mjl18334163602@126.com (J.M.); lx15870181375@126.com (X.L.); wzwang@gzu.edu.cn (W.W.); 2Institute of Crop Protection, College of Agriculture, Guizhou University, Guiyang 550025, China

**Keywords:** kiwifruit canker, biological control, bacterial disease, *Bacillus velezensis*, *Pseudomonas syringae* pv. *actinidiae*

## Abstract

Kiwifruit canker disease, caused by *Pseudomonas syringae* pv. *actinidiae* (Psa), is the main threat to kiwifruit production worldwide. Currently, there is no safe and effective disease prevention method; therefore, biological control technologies are being explored for Psa. In this study, *Bacillus velezensis* WL–23 was isolated from the leaf microbial community of kiwifruit and used to control kiwifruit cankers. Indoor confrontation experiments showed that both WL–23 and its aseptic filtrate had excellent inhibitory activity against the main fungal and bacterial pathogens of kiwifruit. Changes in OD_600_, relative conductivity, alkaline proteinase, and nucleic acid content were recorded during Psa growth after treatment with the aseptic filtrate, showing that Psa proliferation was inhibited and the integrity of the cell membrane was destroyed; this was further verified using scanning electron microscopy and transmission electron microscopy. In vivo, WL–23 promoted plant growth, increased plant antioxidant enzyme activity, and reduced canker incidence. Therefore, WL–23 is expected to become a biological control agent due to its great potential to contribute to sustainable agriculture.

## 1. Introduction

Canker disease, caused by *Pseudomonas syringae* pv. *actinidiae* (Psa) infection, is the most serious kiwifruit disease and has severely affected the development of the kiwifruit industry worldwide [1]. The pathogen has strong transmission ability and can spread rapidly in kiwifruit plantations as it is readily carried by insects, pollen, and agricultural activities [2,3]. The trunks of kiwifruit trees provide rich nutrients and breeding sites for microorganisms. Hence, Psa can use tree trunks as an infection site, resulting in systemic infection of fruit trees that leads to flower browning and abscission, leaf necrosis, and branch canker [4]. Owing to the remarkable transmission ability and pathogenicity of Psa, no effective measures are currently available for disease control in the field. As a result, most kiwifruit orchards worldwide have been destroyed; economic losses are substantial, and the livelihood of kiwifruit farmers has been greatly affected [5,6]. Therefore, effective control of kiwifruit canker disease has become an issue of paramount importance for the future development of the kiwifruit industry.

Currently, the prevention and control of kiwifruit canker disease relies mainly on copper preparations and antibiotics [7]. However, some studies have reported that after plants have been infected by bacterial pathogens, the protective effect of copper preparations will be greatly reduced, and when the concentration is used improperly, their application can cause severe plant damage. Although antibiotics can effectively control bacterial disease incidence, their abuse will accelerate the development of bacterial drug resistance and leave residues in kiwifruit, and several European countries have already banned the use of antibiotics [8]. Concomitantly, increasing public awareness regarding food safety has led to widespread acknowledgement of the need for safe and effective plant disease and pest biological control methods. Furthermore, in recent years, beneficial microorganisms have shown good biocontrol potential in the prevention and control of kiwifruit diseases because of their advantages in terms of safety, economy, and the opportunity for sustainable agricultural development. For example, *Lactobacillus plantarum* has been used to control bacterial pathogens; *Bacillus amyloliquefaciens* has been used to control postharvest fungal pathogens; and bacteriophages have been used to manage kiwifruit canker disease [9,10,11].

*Bacillus* spp. can form endospores to resist adverse conditions, which is advantageous for biological control. Additionally, endophytic *Bacillus* spp. can provide many benefits to plants, including the ability to produce a variety of active molecules with antiviral activity to prevent infection by pathogenic microorganisms [12,13,14]. Inducing plants to enter a stage of stronger defense is known as induced systemic resistance, which refers to resistance to pathogenic microorganisms and insects [15]. On the other hand, *Bacillus* spp. can promote the growth of plants directly (siderophore production, nitrogen fixation, and phytohormone production) or indirectly (production of exo-polysaccharides, hydrogen cyanide, and lytic enzymes) without causing harm to the environment [16,17]. The aim of this study was to evaluate the inhibitory activity and mechanism of action of *Bacillus velezensis* WL–23 against Psa. The biocontrol potential of WL–23 was further evaluated with respect to four aspects: plant growth promotion, induction of plant resistance, the spectrum of antimicrobial activity, and in vivo disease control effects. We believe that our study provides a novel, high-quality biocontrol bacterial resource and a theoretical basis for the development of environmentally friendly biological control agents against kiwifruit diseases.

## 2. Results

### 2.1. Identification of the Bacterial Pathogen

Twenty-one bacterial strains were isolated from the diseased kiwifruit branches. After the specific primers Psa F/R and Tac F/R were used for amplification, three bacterial strains were preliminarily identified as *Pseudomonas syringae* pv. *actinidiae* biovar 3 (Psa3) and used to prepare a bacterial suspension for inoculation of kiwifruit seedlings. After seven days, typical symptoms of Psa infection appeared at the inoculation site. Psa was consistently re-isolated from affected kiwifruit seedlings, fulfilling Koch’s postulates. Therefore, we determined that these three strains of bacteria were Psa3, and strain (P–4) was the most pathogenic with the highest incidence rate (100%) and the largest size of lesion (1.24 cm^2^).

### 2.2. Screening and Identification of Psa Antagonistic Strains In Vitro

Seventy-nine strains of endophytic bacteria were isolated from kiwifruit leaf and branch samples, of which only three showed inhibitory activity against P–4. One endophytic bacterium, WL–23, with the strongest bactericide activity, was isolated from leaves. Therefore, WL–23 was selected for further studies. Small ribosomal subunit 16S rDNA (accession number: OP375578), 1430 bp long, and *gyrA* (accession number: OP382458), 981 bp long, were amplified by PCR sequencing, and BLAST was used in NCBI for comparison with the information in GenBank (Appendix A). Our data showed that WL–23 has high homology with *Bacillus velezensis* WLYS23. The similarity of the 16S rDNA sequence was 99.93% (query cover: 100%), and the similarity of the *gyrA* gene sequence was 100% (query cover: 100%). Related strains were selected, and a phylogenetic tree was constructed using the maximum likelihood method. The phylogenetic tree showed that WL–23 and *B. velezensis* WLYS23 were clustered on the same branch, with a bootstrap value of 100% (Figure 1).

### 2.3. Inhibitory Activity against Kiwifruit Pathogens

Compared with the control group, WL–23 and its aseptic filtrate (AF) showed inhibitory activity against five fungal and two bacterial pathogens (Figure 2). The inhibition ratios of WL–23 for the five kinds of fungi were greater than 65%, and the inhibition ratios of the two kinds of bacteria were greater than 70% (Figure 2A), with the inhibition ratio of P–4 reaching 78.25% (Table 1 and Table 2). After AF treatment, the growth of fungal pathogen colonies was severely inhibited; the growth rate and size of the colonies were much lower than those of the control group (Figure 2B); and the inhibition ratio of the five fungi was greater than 65%. Furthermore, there was an obvious inhibition zone around the filter paper soaked in AF, which inhibited the growth of the bacterial pathogens. The inhibition ratios of the two types of bacterial pathogens were greater than 60%, and the inhibition ratio of P–4 was 67.68%.

### 2.4. Effects of Aseptic Filtrate (AF) on Growth Curve of Psa

Growth curves of Psa treated with different concentrations of AF were determined by measuring the optical density at 600 nm. Psa was normal in the control group, entering a logarithmic growth period after 10 h and a stable growth period after 18 h (Figure 3A). Conversely, although the growth of the 25% AF treatment group showed a sigmoidal pattern, the OD_600_ at each growth stage was lower than that of the infected control group. The logarithmic phase was between 10 and 12 h, and the stable phase was not reached until 20 h. Furthermore, in the 50% AF treatment group, the delay period of the growth curve became longer, the OD_600_ was always at a very low level, the logarithmic growth phase occurred between 12 and 14 h, the decline phase appeared at 20 h, and the OD_600_ gradually decreased from 20 to 24 h. As the AF concentration increased, the OD_600_ of Psa at each growth stage decreased, indicating that Psa growth was severely affected.

### 2.5. Effects of AF on Cell Permeability

To further elucidate the mechanism by which WL–23 inhibits Psa, the permeability of Psa cells was measured after AF treatment. The results indicate that the relative conductivity, AKP, and nucleic acid content in the supernatant of the control group were stable and low during the incubation period of 8 h, with almost no change (Figure 3B–D). However, with an increase in AF concentration, the contents of AKP and nucleic acid in the Psa supernatant increased, and the relative conductivity also increased in a similar pattern. These results demonstrate that AF may increase the permeability of Psa cells, leading to leakage of intracellular substances, including enzymes, nucleic acids, and small ions (K^+^, Ca^2+^, and Na^+^).

### 2.6. Effects of AF on Cell Morphology and the Internal Structure of Psa

Scanning electron microscopy (SEM) and transmission electron microscopy (TEM) observations confirmed the negative effects of AF on the Psa cells. In the control group, the Psa cell membranes were complete and intact, the bacteria were rod-shaped, and their internal structures were stable. Conversely, the surface and interior of the Psa cells changed significantly after the addition of AF. Some bacteria appeared to be broken, deformed, shriveled, and adhered to each other. Furthermore, the surface of Psa cells was damaged, the cell wall disappeared, the cell membrane became thinner, and a large number of intracellular substances leaked out (Figure 4). This change was more evident at the higher AF concentrations tested.

### 2.7. Effects of WL–23 on Kiwifruit

#### 2.7.1. Induced Kiwifruit Resistance

The activities of the three antioxidant enzymes in kiwifruit seedlings inoculated with Psa and watered with WL–23 were higher than those observed in seedlings inoculated with Psa or watered with WL–23 only, likely because of the double stimulation of WL–23-induced resistance and Psa infection (Figure 5). On the second day of the experiment, polyphenol oxidase (PPO) and superoxide dismutase (SOD) activities after Psa inoculation were higher than those of WL–23-watered roots (Figure 5B,C), indicating that the early defense mechanisms of plants may be more sensitive to bacterial pathogens than to induction by WL–23. In general, the activities of the three enzymes after root watering with WL–23 were higher than those in the seedlings inoculated with Psa or in the control group (Figure 5A–C). The results showed that WL–23 induced an increase in the activity of several antioxidant enzymes in kiwifruit trees, thus improving the resistance of kiwifruit seedlings to Psa infection.

#### 2.7.2. Promotion of Plant Growth

Our pot experiments showed that WL–23 exerted a strong growth-promoting effect (Figure 6). Thus, after treatment with C (undiluted AF) and D (1 × 10^8^ CFU/mL WL–23 bacterial suspension) nutrient solutions, seedling height increased, the root system was profuse, and the number of fibrous roots was large (Figure 6C–G). As the AF concentration in the nutrient solution decreased, the growth-promoting effect gradually decreased. The effect of the A (25% AF) and B (50% AF) nutrient solutions on the root length and number of fibrous roots of kiwifruit seedlings was not significant (Figure 6E,F), probably because the concentration of AF in nutrient solution A was too low. Therefore, both WL–23 and high AF concentrations promoted seedling growth, allowing seedlings to grow taller, with lateral roots growing more vigorously, and root systems developing such that seedlings could better absorb soil nutrients and promote tree strength, thereby increasing disease resistance.

### 2.8. In Vivo Control of Psa by AF

Six days after the initiation of the experiment, leaves in the control group were severely diseased (Figure 7a), with large necrotic spots on their left and right sides, and showed a disease index of 80% (Table 3). Disease incidence in the WL–23-treated group was lower than that in the uninoculated control group. WL–23 at 1 × 10^8^ CFU/mL had the best control effect, with only a small number of spots appearing on the right side of the leaf (Figure 7e). Therefore, the overall control effect was 72.22%. Consistently, kiwifruit leaves sprayed with undiluted AF also showed fewer disease spots (Figure 7d), with the control reaching 66.67%. Some spots appeared on the right side of the leaf when sprayed with 50% AF; however, the condition was still better than that on the left side of the leaf (Figure 7c), and the control showed 52.77% disease incidence. When the AF concentration was reduced to 25%, the control effect decreased significantly. A large number of spots appeared on the right side of the leaf (Figure 7b), whereas those of the control reached only 36.11%. These results indicate that WL–23 and its AF could effectively control kiwifruit infection by Psa, and the higher the effective concentration of AF, the greater the control rate attained.

## 3. Discussion

The phyllosphere and rhizosphere are important ecological niches for microbial organisms, among which a large number are beneficial to plants [18,19], and plants maintain their healthy state by interacting with these microorganisms [20]. In this study, the biocontrol bacterial strain WL–23, which has high antibacterial activity against Psa, was isolated from kiwifruit leaves. Through morphological observations and DNA sequencing analysis, WL–23 was identified as *B. velezensis*. Studies have shown that *B. velezensis* is one of the most studied *Bacillus* species for agricultural applications and has great potential for biological pest and disease control [21]. However, in most studies on biological control by *Bacillus* spp., only broad-spectrum inhibitory activity against bacterial or fungal pathogens was targeted [22,23,24]. Interestingly, the experimental results reported herein showed that WL–23 had high inhibitory activity against the main bacterial and fungal pathogens in kiwifruit. This indicates that *B. velezensis* WL–23 has good application value in the production of kiwifruit, especially for the control of kiwifruit canker disease.

The cell membrane is a protective structure that provides a permeability barrier for the passage of small ions and, at the same time, maintains the stability of intracellular substances such as nucleic acid, enzymes, and proteins in the cell [25,26]. Alkaline proteinase (AKP) exists between the cell wall and cell membrane, and its content is almost undetectable in normal cells. When the cell wall is destroyed, AKP penetrates the cell [27]. Previous studies have shown that after treatment with antibacterial active substances, the permeability of bacteria increases, leading to the leakage of intracellular substances [28,29]. In this experiment, after aseptic filtrate (AF) was added, the OD_600_ of the bacterial suspension decreased, and the relative conductivity, AKP, and nucleic acid content in the supernatant culture medium also showed an abnormal upward trend, which is consistent with the above conclusions. The results showed that the integrity of the Psa cells was disrupted, which was further verified by scanning electron microscopy (SEM) and transmission electron microscopy (TEM) observations. Specifically, the cell membrane of Psa was completely ruptured, and the intracellular structure collapsed, showing that AF damaged the cell membrane, acted inside the cell, and disrupted the stability of the cellular structure. Therefore, AF produced by WL–23 may damage the Psa cell wall and membrane, severely hindering Psa growth, which has a significant impact on its propagation. However, Psa treated with different concentrations of AF only showed growth delays or inhibition but could not kill them completely. This situation may be caused by the low concentration of antibacterial active substances in AF produced by Psa [30,31].

In kiwifruit production, inducing plant resistance is also considered an effective method to reduce the occurrence of canker [32]. For example, Reglinski et al. [33] showed that acibenzolar-s-methyl can induce changes in the expression of defense genes and phytohormone content in kiwifruit, thereby improving its disease resistance. On the other hand, de Jong et al. [34] found that the combination of *Aureobasidium pullulans* strain CG163 and acibenzolar-s-methyl with induced resistance could reduce the severity of kiwifruit canker disease. According to the induced production of non-pathogenic microorganisms, or the induced microorganisms, plant systemic resistance can be divided into induced systemic resistance (ISR) and systematic acquired resistance (SAR) [35]. Studies have shown that beneficial microbe-induced ISR can cause plants to exhibit a stronger and faster defense response against pathogens by enhancing the activities of defense-related substances and accumulating reactive oxygen species (ROS) [36,37]. However, excessive accumulation of ROS can also damage tissues and cells, so scavenging ROS by polyphenol oxidase (PPO), superoxide dismutase (SOD), and catalase (CAT) is very important for plant defense responses [38]. In this study, the activities of CAT, PPO, and SOD in kiwifruit seedlings gradually increased after inoculation with WL–23. These three enzymes are involved in plant defense systems. CAT promotes the decomposition of H_2_O_2_ into oxygen and water to remove hydrogen peroxide from the body [39]. PPO oxidizes phenolic compounds to generate highly toxic quinine and participates in the defense response of kiwifruit induced by WL–23 [40]. SOD is mainly used to remove superoxide anion free radicals, O_2_^–^, which are harmful to the body [41]. In addition, our results showed that WL–23 significantly promoted the root system and growth of kiwifruit seedlings. Many studies have reported that most *Bacillus* spp. promote growth through various mechanisms, including plant growth hormone synthesis, nitrogen fixation, and ammonia release from nitrogen-containing organic matter [42,43].

To the best of our knowledge, there are few reports on the biocontrol of bacteria in kiwifruit cankers. Furthermore, most of these studies have focused on pathogen identification, drug development, and the breeding of resistant varieties [44,45,46]. In this study, *B. velezensis* was reported to reduce the incidence of kiwifruit cankers for the first time. Evaluation of the biocontrol effect in vivo is a prerequisite for the development of biocontrol products. WL–23 effects in preventing Psa from infecting kiwifruit are manifold, including the induction of an increase in antioxidant enzyme activity, promotion of healthy plant growth, increase in plant resistance to pathogenic bacteria, destruction of Psa cellular structure, and prevention of Psa expansion. These results show that the strain WL–23 of *B. velezensis* has many beneficial effects on kiwifruit seedlings, thus showing great developmental value and application potential in kiwifruit production.

## 4. Materials and Methods

### 4.1. Isolation and Confirmation of the Bacterial Pathogen

Kiwifruit branches with typical canker symptoms were collected from the field, and a small amount of bark was washed with tap water, soaked in 75% alcohol for 1 min, and washed three times with sterile water for 30 s [47]. Treated samples were ground, diluted, and spread on nutrient agar (NA) medium, and single colonies were selected for purification after three days of incubation at 28 °C. Finally, the purified bacteria were inoculated into a nutrient broth (NB) liquid medium and incubated in a shaker (180 rpm) at 28 °C. The primer Psa F/R can specifically amplify a part of the sequence of the *hopZ3* gene in *Pseudomonas syringae* pv. *actinidiae* (Psa), which can be used for rapid detection of Psa [48], and the primer Tac F/R can specifically detect Psa biovar 3 [49] (the most virulent among several biovars of Psa [8]). Therefore, in order to find out the most pathogenic Psa, the specific primers Psa F/R and Tac F/R were used to detect bacteria by polymerase chain reaction (PCR) amplification. A pathogenicity test was performed by injecting the prepared bacterial suspension into the kiwifruit canes. Bacteria identified as Psa were placed in 50% glycerol and stored at –80 °C.

### 4.2. Isolation and Screening of Antagonistic Bacteria In Vitro

The branches and leaves of healthy kiwifruit trees were collected from various kiwifruit orchards in Guizhou, China (106°33′ E, 27°02′ N). Bacteria were isolated and purified according to the method described in Section 4.1. In short, the sterilized samples were ground, diluted, spread on an NA culture dish, and then incubated in an artificial climate box at 28 °C. After 3 days, single colonies were selected for purification for subsequent experiments. The isolated bacteria were tested for inhibition of Psa activity in vitro according to the method described by Halfeld–Vieira et al. [50]. Briefly, a Psa bacterial suspension (1 × 10^8^ CFU/mL) was mixed with NA medium at a volume ratio of 1:100 (*v:v*), 10 mL of NA containing Psa was poured into an 850 mm Petri dish, and then a 6 mm round filter paper soaked in the suspension of antagonistic bacteria was placed in the center of the Petri dish with three replicates and incubated at 28 °C for two days. Filter paper soaked in NB was used as the control. Strains with the highest antibacterial activity against Psa were screened.

### 4.3. Identification of Antagonistic Bacteria

Genomic DNA of the strain with the strongest inhibitory effect on Psa was extracted using the FastPure bacteria DNA Isolation Kit (Vazyme Biotech Co., Ltd., Nanjing, China). The 16S rDNA sequence was amplified using the bacterial universal primers 27F (5′–AGAGTTTGATCCTGGCTCAG–3′) and 1492R (5′–GGTTACCTTGTTACGACTT–3′) [51]; *gyrA*–F (5′–CAGTCAGGAAATGCGTACGTCTT–3′) and *gyrA*–R (5′–CAAGGTAATGCTCCAGGCATTGCT–3′) [52] were used to amplify the *gyrA* gene sequence. The PCR products were sent to Sangon Biotech Co., Ltd. (Shanghai, China) for sequencing. The nucleotide sequences obtained were submitted to the GenBank database to conduct homologous analysis. A polygene phylogenetic tree was constructed using the maximum likelihood (ML) method in MEGA 7.0, with bootstrap values based on 1000 replications [53].

### 4.4. Preparation of WL–23 Bacterial Suspension and Its Aseptic Filtrate (AF)

Antibacterial WL–23 was obtained according to the method in Section 4.2, which was inoculated into NB and incubated at 37 °C for 48 h. The concentration of WL–23 bacterial suspension was adjusted to an optical density of 600 to 0.3 (OD_600_ = 0.3 approximately, 1 × 10^8^ CFU/mL) with sterile distilled water [54]. Then, WL–23 bacterial suspension (1 × 10^8^ CFU/mL) was inoculated into fresh NB at a volume ratio of 1:100 (*v:v*) and incubated in a shaker (180 rpm) at 37 °C for 5 d. Subsequently, the bacterial suspension was centrifuged at 12,000× *g* for 10 min at 4 °C, and the supernatant was filtered twice through the 0.22 μm pore-size filters to obtain AF [55].

### 4.5. Inhibition Spectrum of the Antagonistic Bacterium and Its AF

In this study, five main fungal and two bacterial pathogens of kiwifruit were selected for testing (Table 4). All strains were stored at the Research Center for Engineering Technology of Kiwifruit, Guizhou University, China. The ability to inhibit bacterial pathogens was tested according to the method described in Section 4.2, and different methods were used for fungal pathogens [56]. The fungal pathogens were prepared as 6 mm mycelial disks with a hole punch and placed in the center of an 850 mm Petri dish. Then, four 6 mm filter papers soaked in the suspension of antagonistic bacteria were evenly placed in four directions, 25 mm away from the mycelial disks, and incubated at 28 °C for three days to measure the inhibition ratio. Three replicates were performed for each species. Petri dishes inoculated with mycelial disks were used as the controls.

The inhibitory effect of the AF on fungal pathogens was evaluated according to the method described by Zheng et al. [64]. Mycelial disks were inoculated at the center of a potato dextrose agar (PDA) plate containing 25% AF (sterile water and AF mixed in a volume ratio of 4:1). The same amount of fermentation medium was added to the PDA medium as a control. The plates were incubated at 28 °C for three days. The inhibition ratio was determined as previously described. The inhibitory activity of AF against bacterial pathogens was measured by slightly modifying the method described by Wu et al. [65]. In this assay, 6 mm filter papers soaked with AF were placed in the center of NA plates containing bacterial pathogens. In the control group, the filter paper was soaked in the fermentation medium. Three replicates were prepared for each species, and the inhibition ratio was measured after three days of incubation at 28 °C. The inhibitory activities of WL–23 and its AF against bacterial pathogens were evaluated by the inhibition zone diameter, and the inhibition ratios of fungi were calculated according to the method described by Cui et al. [66].

### 4.6. Effects of AF on Psa Growth

The method described by Shu et al. [67] was used to measure the growth curve of Psa after treatment to determine the antibacterial activity of AF. With a few modifications, the pathogen Psa was inoculated into fresh NB at a volume ratio of 1:100 (*v:v*), and then an appropriate amount of AF was added to obtain the final content percentages of 25% and 50%, respectively, and incubated in a shaker (180 rpm) at 28 °C for 36 h. In the control group, Psa was inoculated into NB containing the same amount of fermentation medium. During this period, the absorbance of the 2 μL sample at a wavelength of 600 nm was measured every 4 h. Three replicates were used for each concentration. Finally, the growth curve of Psa after AF treatment was plotted with time as the abscissa and the OD_600_ value as the ordinate.

### 4.7. Cell Permeability

The effect of AF on cell permeability of Psa was expressed by relative conductivity, alkaline proteinase (AKP), and nucleic acid content in the Psa supernatant culture medium. Psa was inoculated into NB and incubated overnight. When the OD_600_ reached 0.6, the collected bacteria were washed three times with sterile water. Furthermore, a portion of the collected Psa was washed with a 5% glucose solution until the conductivity of Psa was close to that of a 5% glucose solution. Subsequently, Psa treated by two different methods was inoculated into NB containing 25% and 50% AF, respectively, and incubated in a shaker (180 rpm) at 28 °C for 8 h, with three replicates for each concentration. Psa incubated in NB containing the same amount of fermentation medium was used as the control. The supernatant was collected at 1 h intervals, and AKP activity was detected using the AKP Assay Kit (Jiancheng Bioengineering Institute, Jiangsu, China). The dissolution of nucleic acids was determined at an absorbance wavelength of 260 nm [68]. Relative conductivity was calculated using Equation (1) [69]:(1)$$\mathrm{Relative}\;\mathrm{conductivity}\;\left(\%\right)=(E_{L}-E_{Y})\times 100/E_{Q}$$

*E_L_* is the conductivity of the supernatant, *E_Y_* is the conductivity of the mixture containing a certain concentration of AF in a 5% glucose solution, and *E_Q_* is the conductivity of the Psa mixed with a 5% glucose solution and heated in boiling water for 5 min.

### 4.8. Effects of AF on Cell Morphology and Internal Structure of Psa

Scanning electron microscopy (SEM) and transmission electron microscopy (TEM) were used to observe changes in cell morphology and cell structure [70,71], respectively. Briefly, 50 mL of Psa was centrifuged at 10,000× *g* at 4 °C for 5 min. The collected Psa was resuspended in 100 mL of NB with 25% and 50% AF, respectively, and incubated for 12 h at 28 °C in a shaker at 180 rpm. Psa incubated in NB containing the same amount of fermentation medium was used as the control. The bacterial suspension was centrifuged at 12,000× *g* for 10 min at 4 °C to collect the bacterial cells, rinsed three times with 0.1 mol/L phosphate buffer saline (PBS, pH 7.2), 2.5% glutaraldehyde was added, and the cells were fixed in a refrigerator at 4 °C for 24 h. Samples were eluted with 30%, 50%, 70%, 80%, and 90% ethanol for 15 min at each step and then dehydrated with 100% ethanol for two times, 15 min each time. The treated samples were precooled at −20 ℃ and freeze-dried for 12 h. After conductive coating, the morphology of the Psa cells was observed using SEM (SU8010, Hitachi, Tokyo, Japan), and images were collected. The remaining samples were soaked and embedded with acetone and resin, cut into approximately 100 nm-thick slices on ultra-thin slides, stained with 5% phosphotungstic acid, and then the cell structure of Psa was observed by TEM (HT7800, Hitachi, Tokyo, Japan), and images were collected.

### 4.9. Effects of WL–23 on Kiwifruit

#### 4.9.1. Induced Kiwifruit Resistance

Kiwifruit seedlings were subjected to the following four treatments: (A) roots were watered with WL–23 bacterial suspension after inoculation with Psa screened in Section 4.1; (B) roots were watered with only WL–23 bacterial suspension; (C) inoculation with Psa only; (D) no inoculation, roots not watered. The treated seedlings were maintained at 25–27 °C and 75% relative humidity. Three replicates were prepared per treatment, with 10 kiwifruit seedlings for each treatment. At 0, 2, 6, 10, 15, and 20 days after treatment, 1 g of stem tissue was added to the grinding tube, and 5 mL of precooled 0.05 mol/L PBS (pH 7.2) containing 1.33 mmol/L ethylene diamine tetra acetic acid (EDTA) and 1% polyvinyl polypyrrolidone was added for grinding at 4 °C and 60 Hz for 5 min [72]. The supernatant was collected by centrifugation to detect the enzyme activity.

The catalase (CAT) activity was determined according to a previously described method [73]. Briefly, 0.2 mL of supernatant was mixed with 2.8 mL of H_2_O_2_, and the change in OD_240_ was measured every 30 s for 3 min. A decrease of 0.01 per minute in A_240_ was defined as the enzyme activity unit (U/g FW).

In turn, polyphenol oxidase (PPO) activity was assayed using the method described by Mohammadi and Kazemi [74]. Briefly, 10 μL of the supernatant was added to 290 μL of 50 mmol/L PBS (pH 7.2) containing 100 μmol/L catechol, incubated at 30 °C for 5 min, and OD_398_ was measured every 30 s for 3 min. A decrease of 0.01 per minute in OD_398_ was defined as the enzyme activity unit (U/g FW).

Superoxide dismutase (SOD) activity was determined using a previously described method [75]. Briefly, 4.05 mL of 50 mmol/L PBS (pH 7.2) containing 0.1 mmol/L EDTA, 0.3 mL of 220 mmol/L methionine, 0.3 mL of 1.25 mmol/L nitro-blue tetrazolium, 0.3 mL of 0.033 mmol/L riboflavin, and 0.05 mL of supernatant were mixed. After irradiating the sample with two 15 W fluorescent lamps for 15 min, A_560_ was measured, and the unirradiated sample was used as a blank. The SOD enzyme activity unit was defined as the enzyme activity (U/g FW) responsible for 50% inhibition of the nitro-blue tetrazolium photochemical reduction.

#### 4.9.2. Promotion of Plant Growth

Kiwifruit seedlings of similar height, root length, and fibrous root numbers were selected for pot experiments. Nutrient solutions of the following four components were prepared: (A) 25% AF; (B) 50% AF; (C) undiluted AF; and (D) 1 × 10^8^ CFU/mL of WL–23 bacterial suspension. The kiwifruit seedlings in the treatment groups were watered with 300 mL of nutrient solutions containing these four components, and the controls were watered with the fermentation medium. Three replicates were prepared for each treatment, with five seedlings per replicate. Plant height, root length, and fibrous root number were recorded after cultivation in a greenhouse for 90 days at 26–28 °C and 70% relative humidity (RH).

### 4.10. In Vivo Prevention Assay

The surfaces of the healthy kiwifruit leaves sampled were cleaned with sterile water, wiped with 75% ethanol five times, and finally wiped with sterile water three times [76]. After air-drying on a sterile operating table, a 1 × 10^8^ CFU/mL Psa bacterial suspension was evenly sprayed on the surface of the sterilized kiwifruit leaves. After spraying Psa on the leaf surface for 12 h, a baffle was used to cover the left side of the leaf with the main vein of the leaf as the center, and the four solutions in Section 4.9.2 were evenly sprayed on the right side of the leaf, and then the fermentation medium was evenly sprayed on the left side of the leaf. The fermentation medium was sprayed on the whole leaf surface as a control, with three replicates for each solution and three leaves per treatment. Treated leaves were placed into sterile crispers and cultured in an artificial climate incubator at 16 °C and 75% RH for six days. The extent of leaf symptom occurrence was scored between 0 and 5 according to the ratio of the leaf area occupied by the disease spot on the right side of the leaf compared to the area of the disease spot on the left side of the leaf [77], as follows: 0 = 0%, 1 = 1–20%, 2 = 21–40%, 3 = 41–60%, 4 = 61–80%, and 5 = 81–100%. The disease index and prevention effects were calculated using Equations (2) and (3) [78], respectively:(2)$$\mathrm{Disease}\;\mathrm{index}=\frac{\sum^{\;}(N\times R)}{T\times highhest\;level\left(5\right)}\times 100$$

*N* is the number of leaves showing each disease grade for each treatment, *R* is the representative value of each disease grade, and *T* is the total number of leaves included in the test.
(3)$$\mathrm{Control}\;\mathrm{effect}\;\left(\%\right)=\frac{CK-M}{CK}\times 100$$

*CK* is the disease index of the control, and *M* is the disease index after treatment.

### 4.11. Statistical Analysis

Statistical analyses were conducted using SPSS 22.0 (IBM SPSS, Somers, NY, USA) and Microsoft Excel 2021. All data were checked for normality and equality of variances prior to statistical analysis. One-way analysis of variance (ANOVA) followed by the least significant difference (LSD) test were used to compare the control effect of kiwifruit leaves and the inhibitory effect of fungal and bacterial pathogens after treatment with WL–23 and its AF. Statistical significance was set at *p* < 0.05. Graph-making was performed using Origin 2021 software. All experiments were performed with at least three independent replicates, unless otherwise stated.

## Figures and Tables

**Figure 1 ijms-24-11541-f001:**
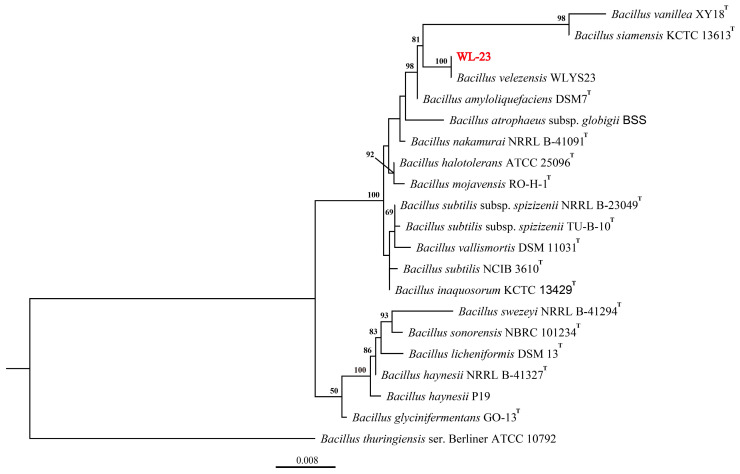
Phylogenetic tree of *Bacillus velezensis* WL–23 based on 16S rDNA and *gyrA* tandem gene sequences, constructed using the maximum likelihood method with MEGA 7.0 software. Bootstrap values are based on 1000 repeats. T = type strain.

**Figure 2 ijms-24-11541-f002:**
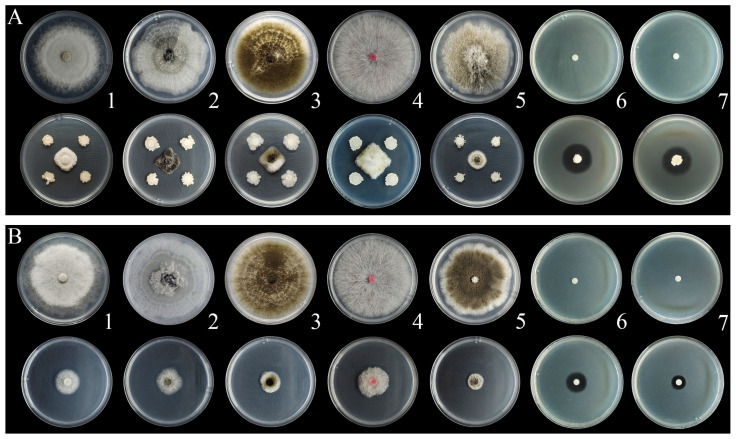
Inhibitory effect of WL–23 (**A**) and 25% aseptic filtrate (AF) (**B**) on kiwifruit pathogens. *Phomopsis longicolla* (ZK–15) (1), *Botryosphaeria dothidea* (ML–31) (2), *Alternaria alternata* (ZK–9) (3), *Fusarium graminearum* (ZK–4) (4), *Didymella glomerata* (LPS-24) (5), *Pseudomonas syringae* pv. *actinidiae* (P–4) (6), and *Agrobacterium fabacearum* (ZK-X-5) (7). Note: The top row of each group represents the control group, and the bottom row represents the treatment group.

**Figure 3 ijms-24-11541-f003:**
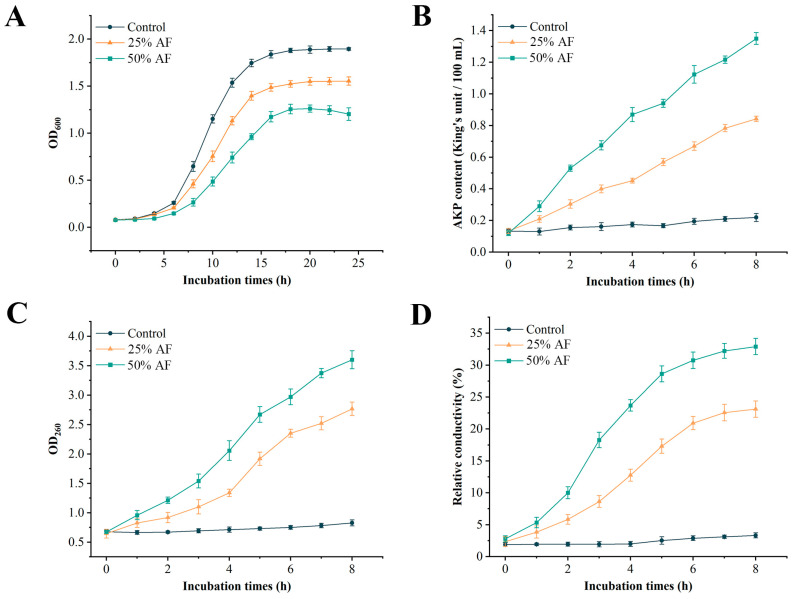
Dynamic changes in Psa growth curve (**A**), AKP content (**B**), nucleic acid content (**C**), and relative conductivity (**D**) after treatment with two different concentrations of AF. Values represent means of triplicates, and error bars represent standard error (SE) (n = 3).

**Figure 4 ijms-24-11541-f004:**
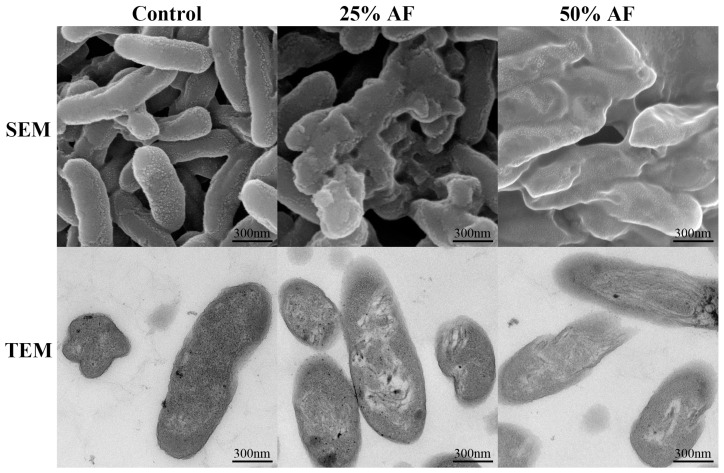
Photography of Psa incubated in 100 mL fermentation medium (first column), 25% AF (second column), and 50% AF (third column) after 12 h by scanning electron microscope (SEM) and transmission electron microscopy (TEM).

**Figure 5 ijms-24-11541-f005:**
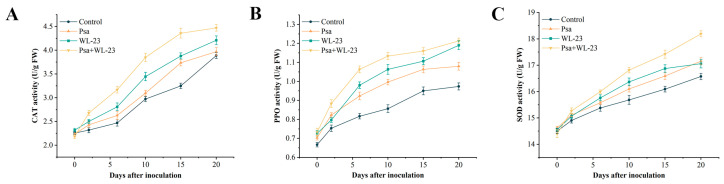
Catalase (CAT) (**A**), polyphenol oxidase (PPO) (**B**), and Superoxide dismutase (SOD) (**C**) activities in kiwifruit seedlings after four different methods of treatment. Values represent means of triplicates, and error bars represents standard error (SE) (n = 3).

**Figure 6 ijms-24-11541-f006:**
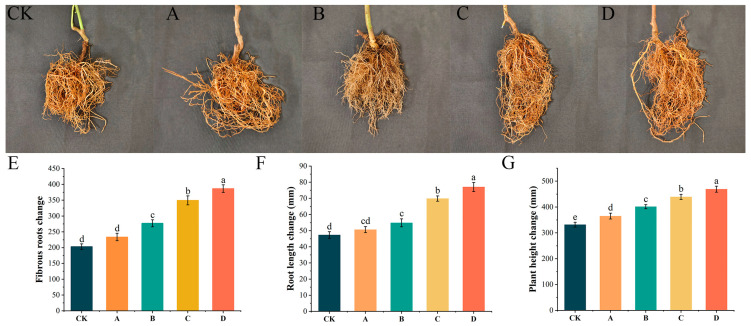
Effects of root watering with sterile water (CK), 25% AF (**A**), 50% AF (**B**), undiluted AF (**C**), and 1 × 10^8^ CFU/mL WL–23 bacterial suspension (**D**) for 90 days on the growth of kiwifruit seedlings and the changes in fibrous roots (**E**), root length (**F**), and plant height (**G**) before and after treatment. Values represent the means of triplicates, and error bars represents standard error (SE) (*p* < 0.05, n = 3).

**Figure 7 ijms-24-11541-f007:**
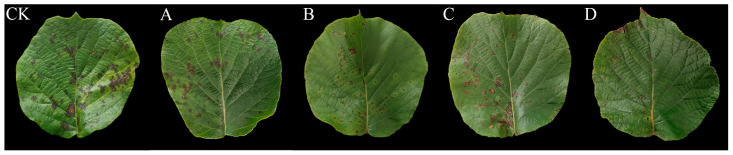
Efficacy of WL–23 to control Psa in vitro. Infected kiwifruit leaves were treated with sterile water (CK), 25% AF (**A**), 50% AF (**B**), undiluted AF (**C**), or 1 × 10^8^ CFU/mL WL–23 bacterial suspension (**D**).

**Table 1 ijms-24-11541-t001:** Inhibitory ratio of WL–23 and its aseptic filtrate (AF) against fungal pathogens.

Fungal Pathogens	Inhibition Ratio by WL–23 (%)	Inhibition Ratio by AF (%)
*Phomopsis longicolla*	74.04 ± 0.55 bc	78.87 ± 0.52 b
*Botryosphaeria dothidea*	71.83 ± 0.76 c	73.33 ± 0.56 d
*Alternaria alternata*	75.02 ± 0.41 b	76.18 ± 0.72 c
*Fusarium graminearum*	66.62 ± 1.10 d	69.40 ± 1.02 e
*Didymella glomerata*	79.76 ± 0.61 a	81.96 ± 0.32 a

Numerical values were expressed as mean ± standard error (SE) of triplicates. Different lowercase letters represent a significant difference in the same columns (*p* < 0.05, n = 3).

**Table 2 ijms-24-11541-t002:** Inhibitory activities of WL–23 and its AF against bacterial pathogens.

Bacterial Pathogens	Inhibition Zone Diameter by WL–23 (mm)	Inhibition Zone Diameter by AF (mm)
*Pseudomonas syringae* pv. *actinidiae*	27.6 ± 0.44 a	18.58 ± 0.43 a
*Agrobacterium fabacearum*	22.54 ± 0.48 b	16.02 ± 0.30 b

Numerical values were expressed as mean ± standard error (SE) of triplicates. Different lowercase letters represent a significant difference in the same columns (*p* < 0.05, n = 3).

**Table 3 ijms-24-11541-t003:** Disease index and control effect of kiwifruit leaves after treatment with WL–23 and its AF.

Treatment	Disease Index	Control Effect (%)
Control	80.00 ± 3.85 a	–
25% AF	51.11 ± 4.44 b	36.11 ± 5.55 c
50% AF	37.78 ± 5.88 c	52.77 ± 7.35 b
Undiluted	26.67 ± 3.85 cd	66.67 ± 4.81 ab
WL–23	22.22 ± 2.22 d	72.22 ± 2.78 a

Numerical values are expressed as the mean ± standard error (SE) of triplicate experiments. Different lowercase letters represent significant differences between the columns (*p* < 0.05, n = 3).

**Table 4 ijms-24-11541-t004:** The pathogens used in antifungal and antibacterial testing.

Pathogens	Disease Type
*Phomopsis longicolla*	Kiwifruit postharvest rot [57]
*Botryosphaeria dothidea*	Kiwifruit soft rot [58]
*Alternaria alternata*	Kiwifruit postharvest rot [59]
*Fusarium graminearum*	Kiwifruit brown leaf spot [60]
*Didymella glomerata*	Kiwifruit black spot [61]
*Pseudomonas syringae* pv. *actinidiae*	Kiwifruit canker [62]
*Agrobacterium fabacearum*	Kiwifruit crown gall [63]

## Data Availability

The data analyzed in this study are included within the paper.

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
