# Peer review of "Assessment of the Biocontrol Potential of Bacillus velezensis WL–23 against Kiwifruit Canker Caused by Pseudomonas syringae pv. actinidiae"

_ijms, 2023, doi:10.3390/ijms241411541_

Round 1

Reviewer 1 Report

Comments and Suggestions for Authors

This paper presents the isolation of a bacterial strain identified as Bacillus velezensis, which produces a compound which disrupts the membrane of Pseudomonas syringae pv. actinidiae (Psa) resulting in a decrease disease incidence on detached kiwifruit leaves. The metabolite(s) produced by B. velezensis WL-23 are not characterized but seem to result in the death of a proportion of the Psa population. Many strains of B. velezensis produce secondary metabolites which inhibit plant pathogens. This strain of Bacillus seems to delay and reduce Psa population in vitro without entirely killing it. More work on plants needs to be done before assessing the real value of this strain as a biological control agent.

In the results section, I do not understand how the authors have found six bright bands after duplex PCR. If they have only six bands (presumably 1 band per line) then they have the positive control of the duplex PCR as described by Balestra et al. A positive result would be two bands per line, i.e. 12 bands total. Also, this PCR should indicate to which biovar the strain of Psa they are using belong to.

Several biovars of Psa have been described with Psa biovar 3 being the most virulent and the real cause of concern (Vanneste, J. L., Yu, J., Cornish, D. A., Tanner, D. J., Windner, R., Chapman, J. R., Taylor, R. K., Mackay, J., and Dowlut, S. 2013. Identification, virulence and distribution of two biovars of Pseudomonas syringae pv. actinidiae in New Zealand. Plant Dis. 97:708-719.). It would be interesting to know whether the work they have done and the results they are presenting are pertinent to this biovar of Psa.

The authors mentioned that the strain of Psa they have selected is the most virulent, could they give some details? Like lesions number or lesions size?

In 2.2 The authors mentioned that WL-23 is bacteriostatic, but the results presented later indicate that Psa is being killed so WL-23 is a or produces a bacteriocide.

When the authors give the percentage of similarity between the 16S of the Bacillus strain they isolated and strains in GenBank, they should mention the percentage of the query covered.

Paragraph 2.3

AF should be spelled out the first time it appears in the text.

I did not understand the inhibition ratios as presented in the text. Did the authors mean the inhibition radius for the bacterial pathogens and the decrease in growth for the fungi?

Figure 2 the authors should explain what the top row represents and what the bottom row represents.

 Paragraph 2.4 . The authors showed that Psa growth is being delayed and limited by the metabolites produced by WL-23. However, they could not kill all of the Psa. Why this is the case, could be brought up in the discussion.

Figure 4 the authors need to explain in the legend what they used to treat the cells.

In the discussion, I would have liked to see the authors discussing the role of the secondary metabolites which kills some of the Psa cells (but not all of them) and the role of the elicitation in the reduction of disease incidence.  There are several examples of control of Psa with biological control agents or by elicitation (de Jong, H., Reglinski, T., Elmer, P. A., Wurms, K., Vanneste, J. L., Guo, L. F., and Alavi, M. 2019. Integrated use of Aureobasidium pullulans strain CG163 and acibenzolar-S-methyl for management of bacterial canker in kiwifruit. Plants 8:287.; Reglinski, T., Vanneste, J. L., Wurms, K., Gould, E., Spinelli, F., and Rikkerink, E. 2013. Using fundamental knowledge of induced resistance to develop control strategies for bacterial canker of kiwifruit caused by Pseudomonas syringae pv. actinidiae. Frontiers in Plant Science 4. Reglinski, T., Wurms, K., Vanneste, J., Ah Chee, A., Yu, J., Oldham, J., Cornish, D., Cooney, J., Jensen, D., and Trower, T. 2022. Transient Changes in Defence Gene Expression and Phytohormone Content Induced by Acibenzolar-S-Methyl in Glasshouse and Orchard Grown Kiwifruit. Frontiers in Agronomy 3:831172.)

In the materials and methods section. Paragraph 4.1 I am very curious about the trunk injection as a pathogenicity test for Psa. Did the authors mean stem instead of trunk?

In paragraph 4.5, the authors need to clarify what they mean by ratio.

In paragraph 4.10. Why did the authors sterilised the leaves before treatment and inoculation? By removing the natural epiphytic microbiome the authors give an advantage to the Bacillus. I wish the authors would repeat this experiment using non-sterilised leaves.

Also how did the symptoms were assessed? Was it by visual scoring or using a programme which calculates the percentage of a leaf which is necrosed?

The literature cited needs to be revised.

Some cited papers do not seem to belong where they are being cited.  For example

I do not think that reference 2 (Zhang, Z.–Z., Long, Y.–H., Yin, X.–H., Yang, S. Sulfur–induced resistance against Pseudomonas syringae pv. actinidiae via triggering salicylic acid signaling pathway in kiwifruit. Int. J. Mol. Sci. 2021, 22, 12710. doi: 10.3390/ijms222312710) is about the impact of Psa worldwide.

In the second line of the introduction, when talking about transmission of Psa by insects, the references 3 and 4 cited by the authors are not about Psa. The paper about the transmission of Psa by insects is: Donati, I., et al. 2020. Pseudomonas syringae pv. actinidiae: ecology, infection dynamics and disease epidemiology. Microb. Ecol. 80:81-102.

Still in the introduction, the impact of Psa on the world kiwifruit production is not presented in reference (8) but it is in Vanneste, J. L. 2012. Pseudomonas syringae pv. actinidiae (Psa): a threat to the New Zealand and global kiwifruit industry. N. Z. J. Crop Hortic. Sci. 40:265-267.

About copper resistance the authors should cite: Colombi, E et al. 2017. Evolution of copper resistance in the kiwifruit pathogen Pseudomonas syringae pv. actinidiae through acquisition of integrative conjugative elements and plasmids. Environ. Microbiol. 19:819-832.

The same paper by Colombi et al. present the genetic basis for streptomycin resistance in Psa while the paper cited by the authors (Reference 10) extrapolates to Psa what had been found in other bacteria .At the end of the second paragraph of the introduction, the example of biological control agents given by the authors (references 11 to 13) are not example of commercially available biological control agents for Psa. On the other hand, there is a strain of Aureobasidium pullulans which is commercially available for control of Psa Hoyte, S., et al. 2018. Development of a new biocontrol product (AUREO® Gold) for control of Pseudomonas syringae pv. actinidiae in kiwifruit. IOBC-WPRS Bull 133:164-166.  

Comments on the Quality of English Language

The English is generally good, but some minor editing is necessary. The main problem I found is what the authors mean by ratio.

Author Response

List of Responses

Dear Reviewers and Editors,

Thank you for your letter and for the reviewers' comments and suggestions concerning our manuscript entitled " Assessment of the biocontrol potential of Bacillus velezensis WL–23 against kiwifruit canker caused by Pseudomonas syringae pv. actinidiae" (ijms-2490852). The reviewers' comments and suggestions are all valuable and very helpful for revising and improving our manuscript. We have carefully revised the manuscript according to the comments and suggestions, and we hope that the revision could meet with your approval. The corrections are marked in red in the revised manuscript. The point to point responses to reviewers’ comments and suggestions are as follows:

Responds to the reviewer's comments and suggestions:

Comment/suggestion 1: In the results section, I do not understand how the authors have found six bright bands after duplex PCR. If they have only six bands (presumably 1 band per line) then they have the positive control of the duplex PCR as described by Balestra et al. A positive result would be two bands per line, i.e. 12 bands total. Also, this PCR should indicate to which biovar the strain of Psa they are using belong to.

Response: Thank you for your comments. In fact, if the target strain is Psa, there is only one band (311 bp) will appear in the gel electrophoresis map after PCR amplification with primer Psa F/R. And in the new manuscript, we added the specific information that PCR are used to identify which biovar the strain of Psa.

Comment/suggestion 2: Several biovars of Psa have been described with Psa biovar 3 being the most virulent and the real cause of concern (Vanneste, J. L., Yu, J., Cornish, D. A., Tanner, D. J., Windner, R., Chapman, J. R., Taylor, R. K., Mackay, J., and Dowlut, S. 2013. Identification, virulence and distribution of two biovars of Pseudomonas syringae pv. actinidiae in New Zealand. Plant Dis. 97:708-719.). It would be interesting to know whether the work they have done and the results they are presenting are pertinent to this biovar of Psa.

Response: We are grateful for the suggestion. In the past experiments, in order to determine whether the Psa (P-4) we used was Psa biovar 3. We supplemented PCR amplification using the specific primer Tac F/R, which can be used for rapid detection of Psa biovar 3 (Yin et al., 2019). The results showed that P-4 belonged to Psa biovar3.

Reference

Yin, Y.–J., Ni, P., Deng, B., Wang, S.–P., Xu, W.–P., Wang, D.–P. Isolation and characterisation of phages against Pseudomonas syringae pv. actinidiae. Acta. Agr. Scand. B 2019, 69, 199–208. doi: 10.1080/09064710.2018.1526965

Comment/suggestion 3: The authors mentioned that the strain of Psa they have selected is the most virulent, could they give some details? Like lesions number or lesions size?

Response: We are extremely grateful to reviewer for pointing out this problem. In fact, P-4 with the strongest pathogenicity was chosen because it had the highest incidence rate (100%), and the largest size of the lesion (1.24 cm2). We added the reasons for selection to the manuscript.

Comment/suggestion 4: In 2.2 The authors mentioned that WL-23 is bacteriostatic, but the results presented later indicate that Psa is being killed so WL-23 is a or produces a bacteriocide.

Response: Thank you for underlining this deficiency. We use the “bacteriocide” instead of “bacteriostatic” in the new manuscript

Comment/suggestion 5: When the authors give the percentage of similarity between the 16S, of the Bacillus strain they isolated and strains in GenBank, they should mention the percentage of the query covered.

Response: We are grateful for the suggestion. We have supplemented the missing content in the new manuscript.

Comment/suggestion 6: AF should be spelled out the first time it appears in the text.

Response: We are grateful for the suggestion. We have already added the spelling of AF when it first appeared

Comment/suggestion 7: I did not understand the inhibition ratios as presented in the text. Did the authors mean the inhibition radius for the bacterial pathogens and the decrease in growth for the fungi?

Response: Thank you for your comments. We have changed the method of determining the inhibitory effect of WL-23 on bacterial pathogens, and now the inhibitory effect is evaluated by the diameter of bacteriostatic circle. On the other hand, according to Cui et al., the inhibition rate of WL-23 against fungal pathogens was calculated using the following formula:

Inhibition ratio (fungal pathogens, %) = (Diameter of fungal pathogen in control group - Diameter of fungal pathogens in treatment group)/ Diameter of fungal pathogen in control group × 100

Reference

Cui, W.–Y., He, P.–J., Munir, S., He, P.–O., Li, X.–Y., Li, Y.–M., Wu, J.–J., Wu, Y.–X., Yang, L.–J., He, P.–F. Efficacy of plant growth promoting bacteria Bacillus amyloliquefaciens B9601–Y2 for biocontrol of southern corn leaf blight. Biol. Control 2019, 139, 104080. doi: 10.1016/j.biocontrol.2019.104080

Comment/suggestion 8: Figure 2 the authors should explain what the top row represents and what the bottom row represents.

Response: Thanks very much for your good comments and suggestions! In the new manuscript, we explain what the top row and the bottom row represent, respectively.

Comment/suggestion 9: Paragraph 2.4. The authors showed that Psa growth is being delayed and limited by the metabolites produced by WL-23. However, they could not kill all of the Psa. Why this is the case, could be brought up in the discussion.

Response: We deeply appreciate the reviewer’s suggestion. In the discussion, we supplemented this part.

Comment/suggestion 10: Figure 4 the authors need to explain in the legend what they used to treat the cells.

Response: Thank you for your comments. We have added this information in the legend.

Comment/suggestion 11: In the discussion, I would have liked to see the authors discussing the role of the secondary metabolites which kills some of the Psa cells (but not all of them) and the role of the elicitation in the reduction of disease incidence. There are several examples of control of Psa with biological control agents or by elicitation (de Jong, H., Reglinski, T., Elmer, P. A., Wurms, K., Vanneste, J. L., Guo, L. F., and Alavi, M. 2019. Integrated use of Aureobasidium pullulans strain CG163 and acibenzolar-S-methyl for management of bacterial canker in kiwifruit. Plants 8:287.; Reglinski, T., Vanneste, J. L., Wurms, K., Gould, E., Spinelli, F., and Rikkerink, E. 2013. Using fundamental knowledge of induced resistance to develop control strategies for bacterial canker of kiwifruit caused by Pseudomonas syringae pv. actinidiae. Frontiers in Plant Science 4. Reglinski, T., Wurms, K., Vanneste, J., Ah Chee, A., Yu, J., Oldham, J., Cornish, D., Cooney, J., Jensen, D., and Trower, T. 2022. Transient Changes in Defence Gene Expression and Phytohormone Content Induced by Acibenzolar-S-Methyl in Glasshouse and Orchard Grown Kiwifruit. Frontiers in Agronomy 3:831172.)

Response: Thank you most sincerely for your attention to our manuscript! In the discussion part, we added the possible reasons for not killing all Psa cells, and discussed the role of induced resistance in reducing the incidence of diseases with reference to the literature you provided.

Comment/suggestion 12: In the materials and methods section. Paragraph 4.1 I am very curious about the trunk injection as a pathogenicity test for Psa. Did the authors mean stem instead of trunk?

Response: Thank you for underlining this deficiency. In this experiment, we actually used kiwifruit branches. In order to describe this process more clearly, we use the word "canes" instead of "trunk".

Comment/suggestion 13: In paragraph 4.5, the authors need to clarify what they mean by ratio.

Response: Thank you for underlining this deficiency. In the new manuscript, we explained the method of preparing 25% AF.

Comment/suggestion 14: In paragraph 4.10. Why did the authors sterilised the leaves before treatment and inoculation? By removing the natural epiphytic microbiome the authors give an advantage to the Bacillus. I wish the authors would repeat this experiment using non-sterilised leaves.

Response: Thank you for your comment. We designed this experiment according to the method of Pinheiro et al. The reason why we sterilize the kiwifruit leaves first is that the collected leaves may carry other kiwifruit leaf pathogens. If disinfection is not performed, other diseases may occur during the experiment and interfere with the results of the experiment. Your point of view provides us with a good idea, and we will further carry out such experiments in the follow-up research.

Reference

Pinheiro, L. A. M., Pereira, C., Barreal, M. E., Gallego, P. P., Balcão, V. M., Almeida, A. Use of phage Ï•6 to inactivate Pseudomonas syringae pv. actinidiae in kiwifruit plants: in vitro and ex vivo experiments. Appl. Microbiol. Biotechnol. 2020, 104, 1319–1330. https://doi.org/10.1007/s00253-019-10301-7

Comment/suggestion 15: Also how did the symptoms were assessed? Was it by visual scoring or using a programme which calculates the percentage of a leaf which is necrosed?

Response: We are grateful for the suggestion. We appreciate your advice. We measured the lesion area with a ruler and then estimated it.

Comment/suggestion 16: I do not think that reference 2 (Zhang, Z.–Z., Long, Y.–H., Yin, X.–H., Yang, S. Sulfur–induced resistance against Pseudomonas syringae pv. actinidiae via triggering salicylic acid signaling pathway in kiwifruit. Int. J. Mol. Sci. 2021, 22, 12710. doi: 10.3390/ijms222312710) is about the impact of Psa worldwide.

Response: Thank you for underlining this deficiency. We replaced " Zhang, Z.–Z., Long, Y.–H., Yin, X.–H., Yang, S. Sulfur–induced resistance against Pseudomonas syringae pv. actinidiae via triggering salicylic acid signaling pathway in kiwifruit. Int. J. Mol. Sci. 2021, 22, 12710. doi: 10.3390/ijms222312710" with " Cameron, A., Sarojini. Pseudomonas syringae pv. actinidiae: chemical control, resistance mechanisms and possible alternatives. Plant Pathol. 2013, 63, 1–11. doi.org/10.1111/ppa.12066" in the new manuscript.

Comment/suggestion 17: In the second line of the introduction, when talking about transmission of Psa by insects, the references 3 and 4 cited by the authors are not about Psa. The paper about the transmission of Psa by insects is: Donati, I., et al. 2020. Pseudomonas syringae pv. actinidiae: ecology, infection dynamics and disease epidemiology. Microb. Ecol. 80:81-102.

Response: Thank you for underlining this deficiency. In view of this problem, we have made adjustments in the new manuscript.

Comment/suggestion 18: Still in the introduction, the impact of Psa on the world kiwifruit production is not presented in reference (8) but it is in Vanneste, J. L. 2012. Pseudomonas syringae pv. actinidiae (Psa): a threat to the New Zealand and global kiwifruit industry. N. Z. J. Crop Hortic. Sci. 40:265-267.

Response: Thank you for the suggestion.  We have replaced the original reference with this one in the new manuscript

Comment/suggestion 19: About copper resistance the authors should cite: Colombi, E et al. 2017. Evolution of copper resistance in the kiwifruit pathogen Pseudomonas syringae pv. actinidiae through acquisition of integrative conjugative elements and plasmids. Environ. Microbiol. 19:819-832.

Response: Thanks very much for your good comments and suggestions! We have replaced the original reference with this one in the new manuscript

Comment/suggestion 20: The same paper by Colombi et al. present the genetic basis for streptomycin resistance in Psa while the paper cited by the authors (Reference 10) extrapolates to Psa what had been found in other bacteria. At the end of the second paragraph of the introduction, the example of biological control agents given by the authors (references 11 to 13) are not example of 9commercially available biological control agents for Psa. On the other hand, there is a strain of Aureobasidium pullulans which is commercially available for control of Psa Hoyte, S., et al. 2018. Development of a new biocontrol product (AUREO® Gold) for control of Pseudomonas syringae pv. actinidiae in kiwifruit. IOBC-WPRS Bull 133:164-166.

Response: Thank you for underlining this deficiency. We have changed " Furthermore, in recent years, beneficial microorganisms have been widely used in the prevention and control of kiwifruit diseases because of their advantages in terms of safety, economy, and the opportunity for sustainable agricultural development." to " Furthermore, in recent years, beneficial microorganisms have shown good biocontrol potential in the prevention and control of kiwifruit diseases because of their advantages in terms of safety, economy, and the opportunity for sustainable agricultural development.".

We appreciate for reviewers and editors' warm work earnestly, and hope that the correction will meet with approval.

Once again, thank you very much for your comments and suggestions.

Reviewer 2 Report

Comments and Suggestions for Authors

General information:

This is a nice well-structured manuscript fitting the scope of the IJMS journal. All the experiments are properly explained, and the data is well presented. There are some minor issues that I have enlisted in the following section. Generally, I missed some supplementary data that should be included in the manuscript submission. The statistical tests should be better presented and the test assumptions should be confirmed. I would suggest a different analytic approach for the curve analysis, but I consider the presented analysis sufficient for the presented data. The language sometimes is a little strange but overall understandable and minor language corrections which are included in the IJMS editorial process should be sufficient. Overall I recommend this manuscript be accepted after minor revision.

In-text comments:

Line 63: Please rephrase that sentence it is not resistance that is responsible for the release of plant chromones.

Line 68: plant growth promotion, induction of plant resistance, the spectrum of antimicrobial activity, and in vivio disease control effect

Line 96 Please abbreviate serovar ser. to be consistent in nomenclature, capitalize, and do not use italics for Berliner.

Line 102: Please define AF, when the abbreviation appears first time in the text abstract, figure or table caption full name should be given.

Line 114: Figure captions should contain full scientific species names, including the tested strain.

Line 114 Please give the strain numbers used in your study.

Line 121 I appreciate the descriptive presentation of data. Usually, the growth curve data are analyzed by statistical comparison in one time point, which loose biological significance. I understand that tools for growth curve analysis are scarce and those available usually do not deliver. However, if I may suggest, I would recommend the usage of “growthcurver” package from R, which fits growth curves to sigmoidal curve models and can be used to compare biologically significant data. This package is also included in the “opm” package more suitable for larger batches of plates to analyze. I would appreciate if you could statistically compare the obtained data for example using the calculated doubling time.

Line 175 no statistical test is mentioned why then use p. Please fully describe the figure, the figure caption should be sufficient to interpret the presented results.

Line 183: Please explain what A and B nutrient solutions are.

Line 292: Please give the localization of the Orchards.

Line 292: Please describe the isolation process of obtained biocontrol strains.

Line 460: You cannot use the parametric test for non-parametric data – visual scoring system.

Please add to the supplementary data:

Pictures from microscopy, growth promotion, and detached leaf assay.

Original data for figures 3, 5, 6, table 1 and 2 can be exel or csv files to ensure that original data can be accessed and reanalyzed. 

Author Response

List of Responses

Dear Reviewers and Editors,

Thank you for your letter and for the reviewers' comments and suggestions concerning our manuscript entitled " Assessment of the biocontrol potential of Bacillus velezensis WL–23 against kiwifruit canker caused by Pseudomonas syringae pv. actinidiae" (ijms-2490852). The reviewers' comments and suggestions are all valuable and very helpful for revising and improving our manuscript. We have carefully revised the manuscript according to the comments and suggestions, and we hope that the revision could meet with your approval. The corrections are marked in red in the revised manuscript. The point to point responses to reviewers’ comments and suggestions are as follows:

Responds to the reviewer's comments and suggestions:

Comment/suggestion 1: Line 63: Please rephrase that sentence it is not resistance that is responsible for the release of plant chromones.

Response: Thank you for underlining this deficiency.  We use "On the other hand, Bacillus spp. Can promote the growth of plants directly (Siderophore production, nitrofixation, and phytohormone production) or indirectly (produc-tion of exo-polysaccharides, hydrogen cyanide, and rhythmic enzymes) without causing harm to the environment "replaced the original sentence.

Comment/suggestion 2: Line 68: plant growth promotion, induction of plant resistance, the spectrum of antimicrobial activity, and in vivo disease control effect

Response: We are grateful for the suggestion. We have made changes in the manuscript.

Comment/suggestion 3: Line 96 Please abbreviate serovar ser. to be consistent in nomenclature, capitalize, and do not use italics for Berliner.

Response: Thank you for your comment. We have modified it in Figure 1.

Comment/suggestion 4: Line 102: Please define AF, when the abbreviation appears first time in the text abstract, figure or table caption full name should be given.

Response: Thank you for your precious comments and advice. We have added the spelling of AF in the corresponding place.

Comment/suggestion 5: Line 114: Figure captions should contain full scientific species names, including the tested strain.

Response: Thank you for your comment. We have supplemented this part in the new manuscript

Comment/suggestion 6: Line 114 Please give the strain numbers used in your study.

Response: We are grateful for the suggestion. We have supplemented this part in the new manuscript

Comment/suggestion 7: Line 121 I appreciate the descriptive presentation of data. Usually, the growth curve data are analyzed by statistical comparison in one time point, which loose biological significance. I understand that tools for growth curve analysis are scarce and those available usually do not deliver. However, if I may suggest, I would recommend the usage of “growthcurver” package from R, which fits growth curves to sigmoidal curve models and can be used to compare biologically significant data. This package is also included in the “opm” package more suitable for larger batches of plates to analyze. I would appreciate if you could statistically compare the obtained data for example using the calculated doubling time.

Response: Thank you for underlining this deficiency. Through our efforts, we can't use these tools well to complete this work. If we need to show our data more clearly, I can modify it into a bar chart.

Comment/suggestion 8: Line 175 no statistical test is mentioned why then use p. Please fully describe the figure, the figure caption should be sufficient to interpret the presented results.

Response: We are very sorry for the mistake we made. We have deleted the "p" in the manuscript.

Comment/suggestion 9: Line 183: Please explain what A and B nutrient solutions are.

Response: We deeply appreciate the reviewer’s suggestion. We have filled in the missing information in the manuscript

Comment/suggestion 10: Line 292: Please give the localization of the Orchards.

Response: Thank you for your suggestion. In the new manuscript we have supplemented the latitude and longitude of the orchard.

Comment/suggestion 11: Line 292: Please describe the isolation process of obtained biocontrol strains.

Response: We are grateful for the suggestion. We have perfected the content in the new manuscript.

Comment/suggestion 12: Line 460: You cannot use the parametric test for non-parametric data – visual scoring system.

Response: We have supplemented this part in the new manuscript. In the new manuscript we have redescribed the contents of this part.

Comment/suggestion 13: Please add to the supplementary data:

Pictures from microscopy, growth promotion, and detached leaf assay.

Original data for figures 3, 5, 6, table 1 and 2 can be exel or csv files to ensure that original data can be accessed and reanalyzed.

Response: Thank you for your comment. we have re-uploaded our original data (figures 3, 5, 6 and table 1, 2, 3), but for the parts of microscopy, growth promotion, and detached leaf assay, we only counted the data without taking photos for saving.

We appreciate for reviewers and editors' warm work earnestly, and hope that the correction will meet with approval.

Once again, thank you very much for your comments and suggestions.

Round 2

Reviewer 1 Report

Comments and Suggestions for Authors

I was very pleased to see that the authors have improved the quality of their paper and followed the comments and suggestions made by the reviewers.  I have only three further comments/suggestions which I would like to see the authors address before their publication goes to print. They are relatively minor and should be easy to address.

About

 Comment/suggestion 2: Several biovars of Psa have been described with Psa biovar 3 being the most virulent and the real cause of concern (Vanneste, J. L., Yu, J., Cornish, D. A., Tanner, D. J., Windner, R., Chapman, J. R., Taylor, R. K., Mackay, J., and Dowlut, S. 2013. Identification, virulence and distribution of two biovars of Pseudomonas syringae pv. actinidiae in New Zealand. Plant Dis. 97:708-719.). It would be interesting to know whether the work they have done and the results they are presenting are pertinent to this biovar of Psa.

Response: We are grateful for the suggestion. In the past experiments, in order to determine whether the Psa (P-4) we used was Psa biovar 3. We supplemented PCR amplification using the specific primer Tac F/R, which can be used for rapid detection of Psa biovar 3 (Yin et al., 2019). The results showed that P-4 belonged to Psa biovar3.

 Reference

Yin, Y.–J., Ni, P., Deng, B., Wang, S.–P., Xu, W.–P., Wang, D.–P. Isolation and characterisation of phages against Pseudomonas syringae pv. actinidiae. Acta. Agr. Scand. B 2019, 69, 199–208. doi: 10.1080/09064710.2018.1526965

I believe that the primers the authors used were also used in the publication they mention  (Yin et al 2019) but they were developed and first described in Koh HS, Kim GH, Lee YS, Koh YJ, Jung JS. 2014. Molecular characteristics of Pseudomonas syringae pv. actinidiae strains isolated in Korea and a multiplex PCR assay for haplotype differentiation. Plant Pathol J. 30:96–101. doi: 10.5423/PPJ.NT.09.2013.0095. Therefore, they should cite this publication.  

About

            Comment/suggestion 16: I do not think that reference 2 (Zhang, Z.–Z., Long, Y.–H., Yin, X.–H., Yang, S. Sulfur–induced resistance against Pseudomonas syringae pv. actinidiae via triggering salicylic acid signaling pathway in kiwifruit. Int. J. Mol. Sci. 2021, 22, 12710. doi: 10.3390/ijms222312710) is about the impact of Psa worldwide.

Response: Thank you for underlining this deficiency. We replaced " Zhang, Z.–Z., Long, Y.–H., Yin, X.–H., Yang, S. Sulfur–induced resistance against Pseudomonas syringae pv. actinidiae via triggering salicylic acid signaling pathway in kiwifruit. Int. J. Mol. Sci. 2021, 22, 12710. doi: 10.3390/ijms222312710" with " Cameron, A., Sarojini. Pseudomonas syringae pv. actinidiae: chemical control, resistance mechanisms and possible alternatives. Plant Pathol. 2013, 63, 1–11. doi.org/10.1111/ppa.12066" in the new manuscript.

The reference Cameron, A., Sarojini. Pseudomonas syringae pv. actinidiae: chemical control, resistance mechanisms and possible alternatives. Plant Pathol. 2013, 63, 1–11.does not address the impact of Psa worldwide and should be deleted.

In that same paragraph the authors have added: ‘(the most virulent among several biovars of Psa [51]). Reference 51 does not demonstrate that biovar 3 is the most virulent biovar. I am not aware of a publication that demonstrates directly the virulence of biovar 3. That biovar 3 is highly virulent has been deduced from how far it spread in different countries. If needed the authors could cite some of the references already mentioned in their paper. Ref 1 or 8 for example.

Overall, the paper is much improved and the authors have done a good job taking into consideration all the comments made by both initial reviewers.

Author Response

List of Responses

Dear Reviewers and Editors,

Thank you for your letter and for the reviewers' comments and suggestions concerning our manuscript entitled " Assessment of the biocontrol potential of Bacillus velezensis WL–23 against kiwifruit canker caused by Pseudomonas syringae pv. actinidiae" (ijms-2490852). The reviewers' comments and suggestions are all valuable and very helpful for revising and improving our manuscript. We have carefully revised the manuscript according to the comments and suggestions, and we hope that the revision could meet with your approval. The corrections are marked in red in the revised manuscript. The point to point responses to reviewers’ comments and suggestions are as follows:

Responds to the reviewer's comments and suggestions:

Comment/suggestion 1:

About

Comment/suggestion 2: Several biovars of Psa have been described with Psa biovar 3 being the most virulent and the real cause of concern (Vanneste, J. L., Yu, J., Cornish, D. A., Tanner, D. J., Windner, R., Chapman, J. R., Taylor, R. K., Mackay, J., and Dowlut, S. 2013. Identification, virulence and distribution of two biovars of Pseudomonas syringae pv. actinidiae in New Zealand. Plant Dis. 97:708-719.). It would be interesting to know whether the work they have done and the results they are presenting are pertinent to this biovar of Psa.

Response: We are grateful for the suggestion. In the past experiments, in order to determine whether the Psa (P-4) we used was Psa biovar 3. We supplemented PCR amplification using the specific primer Tac F/R, which can be used for rapid detection of Psa biovar 3 (Yin et al., 2019). The results showed that P-4 belonged to Psa biovar3.

Reference

Yin, Y.–J., Ni, P., Deng, B., Wang, S.–P., Xu, W.–P., Wang, D.–P. Isolation and characterisation of phages against Pseudomonas syringae pv. actinidiae. Acta. Agr. Scand. B 2019, 69, 199–208. doi: 10.1080/09064710.2018.1526965

I believe that the primers the authors used were also used in the publication they mention  (Yin et al 2019) but they were developed and first described in Koh HS, Kim GH, Lee YS, Koh YJ, Jung JS. 2014. Molecular characteristics of Pseudomonas syringae pv. actinidiae strains isolated in Korea and a multiplex PCR assay for haplotype differentiation. Plant Pathol J. 30:96–101. doi: 10.5423/PPJ.NT.09.2013.0095. Therefore, they should cite this publication.

Response: Thank you for pointing out this problem. In the new manuscript we used "Koh, H. S., Kim, G. H., Lee, Y. S., Koh, Y. J., Jung, J. S. Molecular characteristics of Pseudomonas syringae pv. actinidiae strains isolated in Korea and a multiplex PCR assay for haplotype differentiation. Plant Pathol. J. 2014, 30, 96–101. doi: 10.5423/PPJ.NT.09.2013.0095" to replace the original reference

Comment/suggestion 2:

About

Comment/suggestion 16: I do not think that reference 2 (Zhang, Z.–Z., Long, Y.–H., Yin, X.–H., Yang, S. Sulfur–induced resistance against Pseudomonas syringae pv. actinidiae via triggering salicylic acid signaling pathway in kiwifruit. Int. J. Mol. Sci. 2021, 22, 12710. doi: 10.3390/ijms222312710) is about the impact of Psa worldwide.

Response: Thank you for underlining this deficiency. We replaced " Zhang, Z.–Z., Long, Y.–H., Yin, X.–H., Yang, S. Sulfur–induced resistance against Pseudomonas syringae pv. actinidiae via triggering salicylic acid signaling pathway in kiwifruit. Int. J. Mol. Sci. 2021, 22, 12710. doi: 10.3390/ijms222312710" with " Cameron, A., Sarojini. Pseudomonas syringae pv. actinidiae: chemical control, resistance mechanisms and possible alternatives. Plant Pathol. 2013, 63, 1–11. doi.org/10.1111/ppa.12066" in the new manuscript.

The reference Cameron, A., Sarojini. Pseudomonas syringae pv. actinidiae: chemical control, resistance mechanisms and possible alternatives. Plant Pathol. 2013, 63, 1–11.does not address the impact of Psa worldwide and should be deleted.

In that same paragraph the authors have added: ‘(the most virulent among several biovars of Psa [51]). Reference 51 does not demonstrate that biovar 3 is the most virulent biovar. I am not aware of a publication that demonstrates directly the virulence of biovar 3. That biovar 3 is highly virulent has been deduced from how far it spread in different countries. If needed the authors could cite some of the references already mentioned in their paper. Ref 1 or 8 for example.

Response: Thank you very much for your advice. In the new manuscript we have deleted the reference "Cameron, A., Sarojini. Pseudomonas syringae pv. actinidiae: chemical control, resistance mechanisms and possible alternatives. Plant Pathol. 2013, 63, 1–11. doi.org/10.1111/ppa.12066 ", and replaced reference [51] with reference [8].

We appreciate for reviewers and editors' warm work earnestly, and hope that the correction will meet with approval.

Once again, thank you very much for your comments and suggestions.

Reviewer 2 Report

Comments and Suggestions for Authors

Thank you for appropriately responding to my comments. I do not have any further suggestions and consider this manuscript acceptable for publication in IJMS.

Author Response

List of Responses

Dear Reviewers and Editors,

Thank you for your letter and for the reviewers' comments and suggestions concerning our manuscript entitled " Assessment of the biocontrol potential of Bacillus velezensis WL–23 against kiwifruit canker caused by Pseudomonas syringae pv. actinidiae" (ijms-2490852).

We appreciate for reviewers and editors' warm work earnestly, and hope that the correction will meet with approval.

Once again, thank you very much for your comments and suggestions.